# Repercussions of Bisphenol A on the Physiology of Human Osteoblasts

**DOI:** 10.3390/ijms23105349

**Published:** 2022-05-11

**Authors:** Enrique García-Recio, Víctor J. Costela-Ruiz, Lucía Melguizo-Rodriguez, Javier Ramos-Torrecillas, Olga García-Martínez, Concepción Ruiz, Elvira de Luna-Bertos

**Affiliations:** 1Biomedical Group (BIO277), Department of Nursing, Faculty of Health Sciences, University of Granada, Avda. Ilustración 60, 18016 Granada, Spain; egr@ugr.es (E.G.-R.); vircoss@ugr.es (V.J.C.-R.); luciamr@ugr.es (L.M.-R.); jrt@ugr.es (J.R.-T.); ogm@ugr.es (O.G.-M.); elviradlb@ugr.es (E.d.L.-B.); 2Institute of Biosanitary Research, ibs.Granada, Avda. de Madrid, 15 Pabellón de Consultas Externas, 2ª Planta, 18012 Granada, Spain; 3Institute of Neuroscience, University of Granada, 18016 Granada, Spain

**Keywords:** Bisphenol A, osteoblast, bone tissue, cellular viability, cellular differentiation

## Abstract

(1) Background: Bisphenol A (BPA) is an endocrine disruptor that is widely present in the environment and exerts adverse effects on various body tissues. The objective of this study was to determine its repercussions on bone tissue by examining its impact on selected functional parameters of human osteoblasts. (2) Methods: Three human osteoblast lines were treated with BPA at doses of 10^−5^, 10^−6^, or 10^−7^ M. At 24 h post-treatment, a dose-dependent inhibition of cell growth, alkaline phosphatase activity, and mineralization was observed. (4) Results: The expression of CD54 and CD80 antigens was increased at doses of 10^−5^ and 10^−6^ M, while the phagocytic capacity and the expression of osteogenic genes (ALP, COL-1, OSC, RUNX2, OSX, BMP-2, and BMP-7) were significantly and dose-dependently reduced in the presence of BPA. (5) Conclusions: According to these findings, BPA exerts adverse effects on osteoblasts by altering their differentiation/maturation and their proliferative and functional capacity, potentially affecting bone health. Given the widespread exposure to this contaminant, further human studies are warranted to determine the long-term risk to bone health posed by BPA.

## 1. Introduction

Bisphenol A (BPA), or 4,4′-(Propane-2,2-diyl) diphenol, is a chemical compound used as a major component in the manufacture of polycarbonate plastics, epoxy resins, and other polymeric materials, as well as in certain paper products. This material is employed in the manufacture of products of daily use in the home, including food containers, bottles, utensils, CDs, pens, and toys, among others. It is also used as a coating for food cans, water tanks, as an adjuvant in dental treatments, or as a material for medical devices. However, it is considered that the main route of entry into the human body for BPA is through food as a result of its contact with food containers or utensils [1,2,3]. BPA is considered to be one of numerous endocrine disruptors, i.e., exogenous chemical substances that can hamper the functioning of hormones by binding to their receptors and/or interfering in their transportation due to their resemblance to the natural hormones [4,5]. In this way, BPA can directly bind to different nuclear receptors, such as steroid and xenobiotic receptors (SXRs) [6,7,8]. In particular, the structural similarity between BPA and 17-β-estradiol (E2) allows it to act on the organism by binding to estrogen receptors α and β (ERα and ERβ). BPA also binds to the non-classical estrogen receptor ERγ [9]. 

In addition, BPA has demonstrated the ability to interact with androgen, glucocorticoid, and thyroid hormone receptors [10]. In terms of pharmacokinetics, after absorption, BPA is rapidly metabolized to several inactive metabolites and free BPA is excreted mainly in the urine. Free BPA has been detected in the urine of adults and children, as well as in the serum of pregnant women, umbilical cord serum, and breast milk [10]. Thus, it has the ability to cross the blood-placental barrier and affect the intrauterine development of embryos and fetuses [11].

BPA is involved in the regulation of cancer cell proliferation, migration, invasion, and apoptosis. It is also implicated in anti-cancer drug resistance through several signaling pathways activated by BPA binding to nuclear and membrane receptors such as ERα/β/γ, androgen receptor, and insulin-like growth factor-1 receptor (IGF-1R), among others [10]. Interaction with these receptors also appears to be responsible for BPA toxicity on the reproductive system, interfering with breast formation, germ cell maturation, or placental adhesion to the endometrium. It could also affect inflammatory and immune responses by disrupting different cell signaling pathways mediated by cytokines and various immune cells, such as T and B lymphocytes, macrophages, mast cells, natural killers, or dendritic cells [12].

BPA is known to act on two bone tissue populations, osteoblasts and osteoclasts. In the case of osteoblasts, the bone-forming cells, BPA has been found to inhibit their proliferative capacity. Thus, the in vitro treatment of osteoblasts of murine origin with BPA inhibited their proliferation and induced apoptosis, alkaline phosphatase (ALP) synthesis, calcium nodule formation, and the expression of RUNX2, Osterix, and β catenin genes [4,13,14]. With regard to osteoclasts, the cells responsible for bone resorption, in vitro BPA treatment has been associated with signs of apoptosis and inhibition of their maturation [14]. Therefore, some studies suggest that BPA exposure can disrupt bone homeostasis in both the adult and the fetus [15]. However, we must point out that the data are not conclusive since there are very few studies in human osteoblasts and the exact mechanism of action of BPA on this cell population is unknown.

The objective of this study was to explore the effect of BPA on selected osteoblast parameters by treating human osteoblasts obtained by primary culture with different BPA doses and determining the effect on their growth, differentiation/maturation, and function.

## 2. Results

### 2.1. Effect of BPA on Growth and Cellular Viability

Figure 1A depicts the results obtained for the proliferative capacity of cultured human osteoblasts after 24 h of treatment with BPA doses of 10^−5^, 10^−6^, or 10^−7^ M, showing an inhibitory effect on their capacity at all doses (*p* < 0.0001). The reduction versus controls ranged between 11.60% and 17.53%, obtaining the greatest effect at the highest dose (10^−5^M). The apoptosis/necrosis assay showed that treatment with BPA increased the percentage of apoptotic cells in a dose-dependent manner, inhibiting the percentage of viable cells. No changes were observed in relation to the percentage of necrotic cells (Figure 1B).

### 2.2. Effect of BPA on Antigenic Profile

The flow cytometry results depicted in Figure 2 show that the expression of CD54 and CD80 antigens was significantly increased versus controls after 24 h treatment with BPA at doses of 10^−5^ and 10^−6^ M but was not modified at the lowest dose. No significant change in the expression of CD86 antigen was observed after treatment with any dose.

### 2.3. Effect of BPA on ALP Synthesis

Data depicted in Figure 3 show that the ALP activity of osteoblasts was significantly reduced after 24 h of treatment with BPA at all doses (10^−5^, 10^−6^, and 10^−7^ M), obtaining the greatest effect at the highest dose.

### 2.4. Effect of BPA on In Vitro Mineralization

Figure 4 depicts the results for nodule mineralization of the osteoblasts obtained at 7, 14, and 21 days after culture with BPA at doses of 10^−5^, 10^−6^, and 10^−7^ M for 24 h. No change in mineralization versus controls was observed at day 7 post-treatment, but there was a significant dose-dependent inhibition of mineralization at days 14 and 21.

### 2.5. Effect of BPA on Phagocytic Capacity

Figure 5 shows that the phagocytic capacity of osteoblasts cultured for 24 h with BPA (10^−5^, 10^−6^, or 10^−7^ M) was significantly inhibited versus controls in a dose-dependent manner.

### 2.6. Effect of BPA on Gene Expression

In general, gene expression of the studied osteogenic markers decreased in human osteoblasts cultured in the presence of BPA for 24 h (Figure 6). Treatment significantly reduced the expression of COL-1, RUNX2, OSX, and BMP-7 at all three doses but only reduced the expression of ALP, OSC, and BMP-2 at the higher doses (10^−5^ and 10^−6^M).

## 3. Discussion

In this in vitro study, treatment with BPA inhibited the proliferative capacity by apoptosis induction and cell differentiation of human osteoblasts obtained by primary culture and modified their antigen and gene expression. Given the key role of osteoblasts in bone physiology, these results suggest that BPA exerts adverse effects on bone, in line with reports of the negative impact of this endocrine disruptor on many other human tissues [16,17,18]. These findings are in line with previous observations in mouse [14] and human fetal [19] osteoblast cell lines. This is the first study to examine these effects in human osteoblasts obtained by primary culture.

BPA demonstrated a dose-dependent inhibitory effect on osteoblast proliferative capacity by inducing apoptosis at doses of 10^−5^, 10^−6^, and 10^−7^ M. Hwang et al., 2013, also observed that BPA at concentrations of 2.5–12.5 µM inhibited the growth of MC3T3-E1 mouse cells by apoptosis, which was stimulated at 24 h by caspase activation; however, they detected no effects on cell viability at a concentration of 0.5 µM, the lowest dose studied. In their study of the human fetal osteoblast cell line hFOB1.19, Thent et al. (2018) reported that a concentration of 12.5 µg/mL BPA was necessary to reduce cell viability by 50% at 24 h. Cadmium, another endocrine disruptor, also demonstrated an anti-proliferative effect on human osteoblasts at 24 h of treatment [20]. Given that osteoblasts are responsible for generating new bone tissue in the complex and continuous process of bone remodeling, inhibition of their proliferative capacity may have severe consequences, potentially leading to the loss of bone mass and density [21,22,23].

The results obtained for ALP activity and mineralization also indicate a negative effect of BPA on bone matrix formation. ALP synthesis increases concentrations of calcium and phosphorus in the bone matrix, and its reduction would be involved in the inhibition of mineralization detected at 14 and 21 days of treatment [24]. Contradictory results have been obtained for the MC3T3-E1 mouse cell line, with one study finding that BPA treatment reduced ALP activity and calcium nodule formation [14] and another that it increased ALP activity and bone mineralization [25]. Mineralization provides bone with rigidity and resistance, and its alteration can therefore have a negative impact on bone density and quality [26].

In the present study, BPA treatment was associated with a decreased expression of CD54 and CD80 surface antigens, which can also be modulated by cytokines, growth factors, platelet-rich plasma, bacterial lipopolysaccharides, phenolic compounds, and certain pharmaceuticals; reduced expression of these markers has been associated with the maturation/differentiation of osteoblasts and their elevated expression with suppression of their differentiation [27,28,29,30]. These alterations in the expression of CD54 and CD80 are consistent with the inhibition by BPA of ALP synthesis and mineralization in osteoblasts, which suggests a suppression of their maturation. BPA was also found to inhibit the phagocytic capacity of osteoblasts, which is known to be altered by their in vitro treatment with non-steroidal anti-inflammatory drugs, laser radiation, or phenolic compounds [27,28,29].

BPA treatment also inhibited the expression of ALP, COL-1, OSC, RUNX2, OSX, BMP-2, and BMP-7, osteogenic markers related to osteoblastogenesis and osteoblast function [31]. The presence of BPA was previously found to inhibit the expression of RUNX2 and OXS and the Wnt/β-catenin signaling pathway in the MC3T3-E1 mouse cell line, suggesting the inhibition of osteoblast differentiation and therefore of bone formation [14]. The reduced expression of RUNX2 and OSX observed in the present study would be related to a change in differentiation, while the reduced expression of ALP, COL-1, OSC, BMP-2, and BMP-7 would indicate a negative effect not only on the maturation of osteoblasts but also on their functional capacity.

Caution should be taken in evaluating the potential clinical relevance of these in vitro results. Nevertheless, similar findings have led to confirmation of the clinical impact of BPA exposure on other body tissues, such as adipose tissue, placenta, sperm, and mammary glands, among others [32,33]. There is an urgent need to develop novel approaches to the prevention and therapy of bone disease, especially osteoporosis, considered a pandemic by the WHO [34] and largely attributed to an imbalance in bone remodeling [35,36]. In this sense, some authors have shown that olive leaf extracts, rich in phenolic compounds such as hydroxytyrosol or oleuropein, have effects against metabolic disorders induced by BPA by improving the antioxidant defense system and regulating important activities of pathways signaling [37]. Hence, further research is warranted to verify the effect of BPA exposure on osteoclasts and osteoblasts, which mediate the resorption and formation of bone, respectively.

## 4. Materials and Methods

### 4.1. Chemical

BPA (C15H16O2) was purchased from Sigma-Aldrich (Co., St. Louis, MO, USA) and dissolved in dimethyl sulfoxide (DMSO); the final concentration of DMSO never exceeded 0.05%.

### 4.2. Primary Human Osteoblasts

Primary human osteoblasts were obtained from bone chips gathered during routine mandibular osteotomy or lower wisdom tooth extraction in healthy patients at the Clinic of the School of Dentistry of the University of Granada. All study procedures were carried out in accordance with the 1964 Helsinki declaration and its later amendments or comparable ethical standards. All participants signed informed consent for participation in the study, which was approved by the Ethical Committee of the University of Granada (Reg. No. 523/CEIH/2018). Bone samples were independently processed, with thorough washing of fragments four times in phosphate-buffered saline (PBS, pH 7.4) to remove bone marrow and periosteum remains. The bone fragments were then seeded into culture flasks and cultured according to a previously reported protocol [38]. Finally, three primary osteoblast cell lines were established.

### 4.3. Treatments

The osteoblast cells obtained were treated for 24 h with BPA at doses of 10^−5^ M, 10^−6^ M, or 10^−7^ M; untreated cells served as controls.

### 4.4. Cell Proliferation

The MTT method was used to determine cell proliferation [39]. Osteoblasts were seeded at 1 × 10^4^ cells/mL per well into a 96-well plate (Falcon, Becton Dickinson Labware, Franklin Lakes, NJ, USA) and cultured at 37 °C in a humidified atmosphere of 95% air and 5% CO2 for 24 h in Dulbecco’s Modified Eagle medium (DMEM) (Invitrogen Gibco Cell Culture Products, Carlsbad, CA) with 20% fetal bovine serum (FBS) (Gibco, Paisley, UK). The medium was then replaced with DMEM containing BPA at doses of 10^−5^, 10^−6^, or 10^−7^ M. At 24 h, the medium was replaced with phenol-red-free DMEM containing 0.5 mg/mL 3-(4,5-dimethylthiazol-2-yl)-2,5-diphenyltetrazolium bromide (MTT) (Sigma, St. Louis, MO, USA) and incubated for 4 h. Cellular reduction of the MTT tetrazolium ring resulted in the formation of a dark-purple water-insoluble deposit of formazan crystals. After incubation, the medium was aspirated and DMSO was added to dissolve the formazan crystals. Absorbance was measured with a spectrophotometer at 570 nm (SunriseTM, Tecan, Männedorf, Switzerland).

### 4.5. Apoptosis and Necrosis Analysis

Cultured human osteoblast cells treated with 10^−5^, 10^−6^, or 10^−7^ M for 24 h and untreated control cells were detached from the culture flask, washed, suspended in 300 µL PBS, and then labeled with annexin V and PI (Immunostep S.L., Salamanca, Spain), incubating 100 µL aliquots of the cell suspension with 5 µL annexin V and 5 µL PI for 15 min at room temperature in the dark. Cells were then washed, suspended in 500 µL PBS, and immediately analyzed in a flow cytometer with argon laser (Facs Vantage Becton Dickinson, Palo Alto, Santa Clara, CA, USA) at a wavelength of 488 nm to determine the percentage of fluorescent cells. We calculated the percentage of annexin-positive (apoptotic) cells and PI-positive (necrotic) cells from counts of 2000–3000 cells [40].

### 4.6. Antigenic Phenotype

The antigenic phenotype was studied by flow cytometry at 24 h of culture with 10^−5^, 10^−6^, or 10^−7^ M BPA; untreated cells served as controls. Cells were detached from the culture flask with 0.4% (*w*/*v*) EDTA solution, washed, and suspended in PBS at 2 × 10^4^ cells/mL. Cells were labeled by direct staining with monoclonal antibodies (MAbs) CD54 (ICAM-1 MAb [MEM-111], FICT), CD80 (human CD80 [B7-1, BB1], FICT) and CD86 (human CD86 [B7-2, B70], FICT) from Invitrogen, Thermo Fisher Scientific, Madrid, Spain. Cells were then analyzed by flow cytometry (FASC Canton II, SE Becton Dickinson, Palo Alto, CA) as described by [27].

### 4.7. ALP Activity

ALP activity was quantified using a colorimetric assay (diagnostic kit 104-LL; Sigma) as described by [29]. Primary osteoblasts cultured in non-osteogenic medium and treated with BPA at the above-reported doses or untreated (controls) were seeded at 1 × 10^4^ cells/mL per well into 24-well plates and cultured for 24 h under standard conditions. Standards of p-nitrophenol (0 to 250 μmol/L) were prepared from dilutions of a 1000 μmol/L stock solution and assayed in parallel. Next, cells were lysed with 0.1% (*v*/*v*) Triton X-100 at 37 °C, and the cell lysate was removed in a known volume of buffer containing 10 mM Tris-HCl pH 7.8 and 0.5 mM MgCl_2_. ALP activity was determined using the p-nitrophenyl phosphate (p-NPP) liquid substrate system (Sigma), which measures the formation of yellow p-NP from p-NPP as catalyzed by ALP. Briefly, 50 μL of cell lysate solution was added to the ALP substrate and incubated at 37 °C for 45 min; next, 50 μL of NaOH 0.1 mol/L was added in each sample to stop the enzymatic reaction, and a final absorbance measurement was taken at 405 nm with an ELx800 spectrophotometer (BioTek, Winooski, VT, USA). Total protein content was estimated by the Bradford method using a protein assay kit from Bio-Rad Laboratories (Nazareth-Eke, Belgium). ALP activity was expressed as U/mg protein. Results of each assay were compared with those for untreated cells grown under the same conditions.

### 4.8. Mineralization Assay

Cells were cultured in DMEM with ascorbic acid (0.05 mM) and β-glycerol phosphate (5 mM). At 7, 14, and 21 days, the plate with cells and the precipitated calcium incorporated in the cell matrix were stained with alizarin red S (2%). The dye present in mineralization nodules was extracted for 15 min with 10% (*w*/*v*) cetylpyridinium chloride in 10 mM sodium phosphate (pH 7.0). The extracted stain was then transferred to a 96-well plate and the absorbance at 562 nm was measured with a spectrophotometer (ELx800, BioTek) as previously described [41].

### 4.9. Phagocytic Activity

Flow cytometry was used to study the phagocytic activity of osteoblasts cultured for 24 h with BPA (10^−5^, 10^−6^, or 10^−7^ M) or without BPA (controls). Cells were detached from the culture flask using a solution of 0.05% (*w*/*v*) trypsin and 0.02% (*w*/*v*) EDTA, washed, and suspended in complete culture medium with 10% (*v*/*v*) FBS at 2 × 10^4^ cells/mL. Cells were labeled by direct staining with labeled latex beads, incubating 100 µL of cell suspension with 2 μL of carboxylated FICT-labeled latex beads with diameter of 2 μm (Sigma Aldrich, St Louis, MO, USA) for 30 min at 37 °C in darkness. Cells were washed, suspended in 1 mL of PBS, and immediately analyzed in a flow cytometer (FASC Canton II, SE Becton Dickinson, Palo Alto, Santa Clara, CA, USA). Results were expressed as the percentage of cells positive for phagocytosis with respect to control group.

### 4.10. Gene Expression Analysis

Gene expression was determined by real-time polymerase chain reaction (RT-qPCR) as described by [42]. After 24 h of culture with or without BPA (controls), cells were detached from the culture flask using 0.05% trypsin-EDTA solution (Sigma) and individually harvested. Messenger RNA (mRNA) was extracted by a silicate gel technique in the QiagenRNeasy extraction kit (Qiagen Inc., Hilden, Germany), which includes a DNAse digestion step. The amount of extracted mRNA was measured by UV spectrophotometry at 260 nm (Eppendorf AG, Hamburg, Germany), determining contamination with proteins according to the 260/280 ratio. An equal amount of RNA (1 μg of total RNA in 40 μL of total volume) was reverse-transcribed to cDNA and amplified by PCR using the iScript™ cDNA Synthesis Kit (Bio-Rad laboratories, Hercules, CA, USA), following the manufacturer’s instructions.

Primers were designed using the NCBI-nucleotide library and Primer design (Table 1) to detect mRNA of ALP, COL-1, OSC, RUNX2, OSX, BMP-2, and BMP-7. All were matched to the mRNA sequences of target genes (NCBI Blast software). Final results were normalized as proposed by Ragni et al. (2013). Quantitative RT-PCR (RT-qPCR) was performed using the SsoFast™ EvaGreen^®^ Supermix Kit (Bio-Rad laboratories) in accordance with the manufacturer’s protocols. Samples were amplified in 96-well microplates in an IQ5-Cycler (Bio-Rad laboratories) using an annealing temperature (specific for each gene) ranging from 60 °C to 65 °C and an elongation temperature of 72 °C over 40 cycles. PCR reactions were carried out in a final volume of 20 μL, with 5 μL of cDNA sample and 2 μL of each primer. For the negative control, water was used instead of mRNA samples. Standard curves were constructed for each target gene by plotting Ct values against log cDNA dilution. After each RT-PCR, a melting profile was created and agarose gel electrophoresis of each sample was carried out to rule out nonspecific PCR products and primer dimers. The relative quantification of gene expression was determined using the comparative Ct method. For each gene, the mRNA concentration was expressed in ng of mRNA per average ng of housekeeping mRNAs. The cDNA from individual cell experiments was analyzed in triplicate RT-PCR studies. The data were expressed as the percentage of expression with respect to control group.

### 4.11. Statistical Analysis

SPSS version 24.0 (IBM SPSS, Armonk, NY, USA) was used for the statistical analysis. Each experiment was performed in triplicate for each culture. Results were expressed as means ± SD, and the Kolmogorov–Smirnov test was applied to evaluate the normality of variable distributions. One-way analysis of variance (ANOVA) was performed, applying the Bonferroni correction. *p* < 0.05 was considered significant. All data were analyzed in relation to the control group.

## 5. Conclusions

These in vitro findings confirm the significant adverse effects of BPA on human osteoblasts. Exposure inhibits their growth, differentiation, and function through decreases in ALP synthesis and mineralization and changes in antigen expression and phagocytic capacity, closely related to alterations in gene expression. These results support the need to prevent the exposure of humans to this toxic pollutant, which is widely present in the environment and the home [32,43]. Our results contribute to explaining the increase in the prevalence of bone pathologies, such as osteoporosis, as a consequence of the high exposure of the population to this endocrine disruptor. However, further in vitro and in vivo research is needed to verify the repercussions of long-term exposure to BPA on bone health.

## Figures and Tables

**Figure 1 ijms-23-05349-f001:**
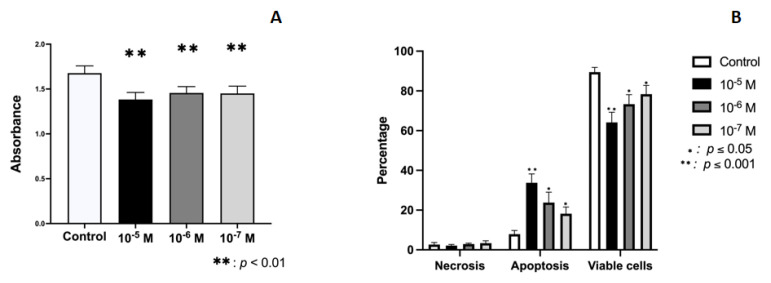
Effect of BPA at different doses (10^−5^, 10^−6^, or 10^−7^ M) on osteoblast growth and cellular viability in primary cell lines after 24 h of incubation. (**A**) Effect on cellular proliferation. (**B**) Effect on induction of apoptosis/necrosis. Data are expressed as means ± standard deviation. Significant differences * *p* < 0.05; ** *p* < 0.001.

**Figure 2 ijms-23-05349-f002:**
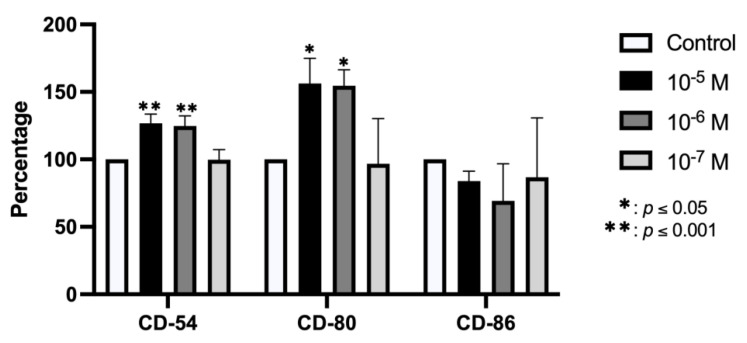
Expression of CD54, CD80, and CD86 in cells treated for 24 h with BPA (10^−5^, 10^−6^, or 10^−7^ M). Data are expressed as percentage expression with respect to control ± standard deviation. Significant differences * *p* < 0.05; ** *p* < 0.001.

**Figure 3 ijms-23-05349-f003:**
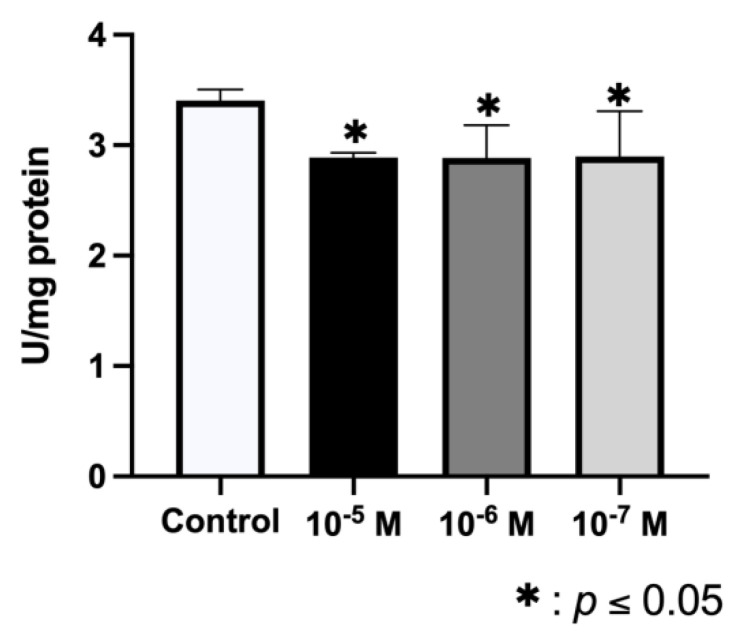
ALP activity of primary cell lines after 24 h of treatment with BPA at different doses of 10^−5^, 10^−6^, or 10^−7^ M. The activity was determined in cell lysates and normalized to total cellular protein (U/mg of proteins). Data are reported as means ± standard deviation. Significant differences * *p* < 0.05.

**Figure 4 ijms-23-05349-f004:**
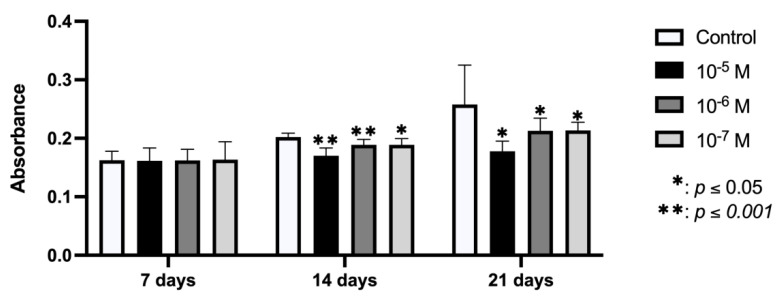
Quantitative study of mineralization of primary cell lines through nodule formation, after culture in osteogenic medium supplemented with BPA (10^−5^, 10^−6^, or 10^−7^ M). Data are reported as means of absorbance ± standard deviation. Significant differences * *p* < 0.05; ** *p* < 0.001.

**Figure 5 ijms-23-05349-f005:**
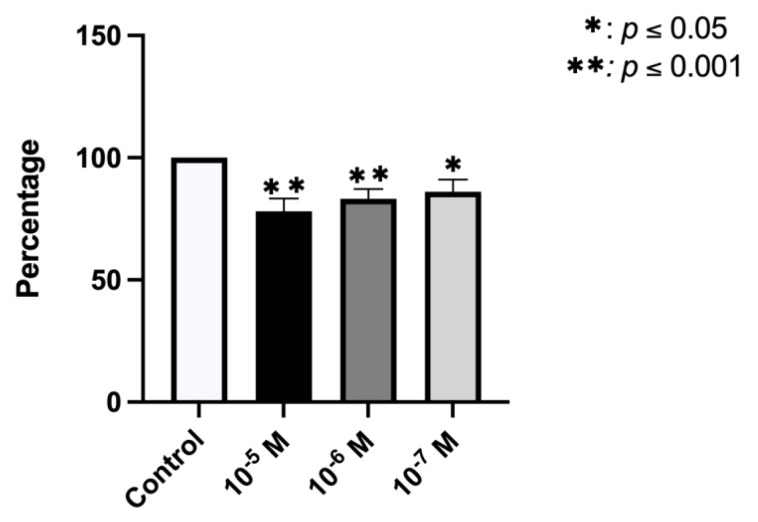
Phagocyte capacity of osteoblasts cell lines after treatment with BPA at different doses of 10^−5^, 10^−6^, or 10^−7^ M determined by means of flow cytometry. The data are expressed as the mean of the percentage of cells positive for phagocytosis with respect to control group ± standard deviation. Significant differences * *p* < 0.05; ** *p* < 0.001.

**Figure 6 ijms-23-05349-f006:**
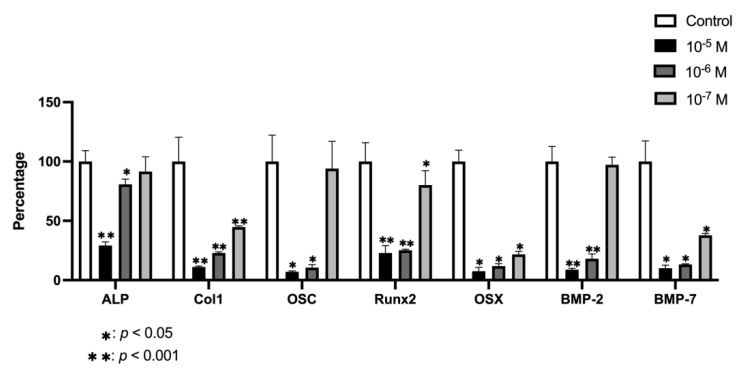
Expression of osteoblast genes (ALP, COL-1, OSC, RUNX2, OSX, BMP-2, and BMP-7) treated for 24 h with BPA at doses of 10^−5^, 10^−6^, or 10^−7^ M. The data are expressed as the mean of expression percentage with respect to control group ± standard deviation. * Significant differences *p* < 0.05, ** *p* < 0.001.

**Table 1 ijms-23-05349-t001:** Primer sequences for the amplification of cDNA by real-time PCR.

Gene	Sense Primer	Antisense Primer	Amplicon (bp)
ALP	5′-CCCATATTCCCTGCACTTTG-3′	5′-ACCTTGACCTCTCAGCCTCA-3′	195
Col-1	5′-CCTCATCGCAGGAGAAAAAG-3′	5′-CCCTGAAGTGACTGGGGTAA-3′	169
OSC	5′-CCTGGTCCAGACCACAGAGT-3′	5′-TGGAGATTTTGGGAGTACGG-3′	194
Runx2	5′-CCTTGCTGCTCTACCTCCAC-3′	5′-CACACAGGATGGCTTGAAGA-3′	197
OSX	5′-TGCCTAGAAGCCCTGAGAAA-3′	5′-TTTAACTTGGGGCCTTGAGA-3′	205
BMP-2	5′-TCGAAATTCCCCGTGACCAG-3′	5′-CCACTTCCACCACGAATCCA-3′	142
BMP-7	5′-CTGGTCTTTGTCTGCAGTGG-3′	5′-GTACCCCTCAACAAGGCTTC-3′	202

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
