# Peer review of "Repercussions of Bisphenol A on the Physiology of Human Osteoblasts"

_ijms, 2022, doi:10.3390/ijms23105349_

Round 1

Reviewer 1 Report

Bisphenol A (BPA) is an endocrine disruptor which can bind to the oestrogen receptor. It also possesses oestrogenic, antiandrogenic, inflammatory and oxidative properties. Since bone responds to changes in sex hormones, inflammatory and oxidative status, BPA exposure could influence bone health in humans.

The advantage here is that you use many complimentary techniques, both cytotox, proliferation, ALP, Mineralization Assay, Gene expression analysis., It also a strength that you have harvested the human osteoblasts yourselfs. However some issues needs to be resolved prior to acceptance.

-Since you are working with human cells, you either need to do the experiments from several donors or in triplicate. I understand that you pooled together human osteoblasts from several biopsies? In this case, you need to do triplicates. which you did-well done! however,  I also need some assurance that 21 d is enough not to induce mineralization. Your control at 21d has a very high SD. Most human osteoblasts studies cultivate their cells until 28 d. For cell lines, 21 d is enough, but needs some evidence that 21 d is enough

-Do I understand correctly that the qPCR was only done after 24hrs of cultivation? This does not make any sense as several of the analyzed gene expression profiles are not appropriately expressed in such a short period. you need to redo the qPCR for 7, 14, and 21 days. It is a little far fetch to conclude that BPA inhibits osteoblast differentiation if your qPCR only is for 24 hrs

But great that you added the primers, most people forget!

-Please make sure that you have the molarities in superscript doses of 10-5, 10-6, or 10-7 M. A not "doses of 10-5, 10-6, or 10-7 M. A"

  • please revise sentence "Finally, three primary osteo-208 blast cell lines were established." I guess cell lines is wrong here since you used primary human cells
  • please state that you Statistical analysis was done with the control as control. However it also be interesting to know if you had sign diff betweent he different BPA concentrations

Author Response

Reviewer 1

Bisphenol A (BPA) is an endocrine disruptor which can bind to the oestrogen receptor. It also possesses oestrogenic, antiandrogenic, inflammatory and oxidative properties. Since bone responds to changes in sex hormones, inflammatory and oxidative status, BPA exposure could influence bone health in humans.

The advantage here is that you use many complimentary techniques, both cytotox, proliferation, ALP, Mineralization Assay, Gene expression analysis., It also a strength that you have harvested the human osteoblasts yourselfs. However some issues needs to be resolved prior to acceptance.

  1. Since you are working with human cells, you either need to do the experiments from several donors or in triplicate. I understand that you pooled together human osteoblasts from several biopsies? In this case, you need to do triplicates. which you did-well done! however,  I also need some assurance that 21 d is enough not to induce mineralization. Your control at 21d has a very high SD. Most human osteoblasts studies cultivate their cells until 28 d. For cell lines, 21 d is enough, but needs some evidence that 21 d is enough

Response: To carry out our study we have followed a classical methodology of cell line establishment by primary culture. We used 3 bone samples from three different patients, and from them we established 3 different human osteoblast cell lines. These lines are studied independently and are not mixed at any time.  Each assay (viability, phosphatase synthesis, mineralization, gene expression, etc.) was performed on all 3 lines and each experiment on each line was performed at least in triplicate.

Response: In the mineralization study, prior to the extraction of the red alizarin that stains the calcium nodules, these are observed under the microscope to make a qualitative assessment. In our study,  clear differences were observed between the control group and the treatment groups. Quantitative analysis by spectrophotometry has the disadvantage of requiring extraction of the red alizarin with cetylpyridinium chloride. This is a delicate process since the dye must be extracted without lifting the cells adhered to the surface of the culture flask, which partly explains the high variability that is observed in these studies, as we can see in the following article:

  • Sabandal et al., Head Face Med. 2020 Aug 20;16(1):18. doi: 10.1186/s13005-020-00232-4.
  • Brennan et al. Horm Metab Res. 2014 Jul;46(8):537-45. doi: 10.1055/s-0033-1363265.

Another factor that influences the variability of the results is that we are working with 3 different cell lines, coming from different donors.

For mineralization assays, 7, 14 and 21 days is the most common time period used by different researchers, as we can see in the following articles:

  • Li et al. Mol Med Rep. 2019 May;19(5):3676-3684. doi: 10.3892/mmr.2019.10040. (7 days)
  • Thent et al.. Life Sci. 2018, 210, 214–223, doi:10.1016/j.lfs.2018.08.057.(6 days)
  • Sabandal et al., Head Face Med. 2020 Aug 20;16(1):18. doi: 10.1186/s13005-020-00232-4. (9 and 13 days)
  • Brennan et al. Horm Metab Res. 2014 Jul;46(8):537-45. doi: 10.1055/s-0033-1363265 (7 and 14 days)
  • Wagner et al. J Bone Miner Metab. 2022 Jan;40(1):20-28. doi: 10.1007/s00774-021-01269-4.(21 days)
  1. Do I understand correctly that the qPCR was only done after 24hrs of cultivation? This does not make any sense as several of the analyzed gene expression profiles are not appropriately expressed in such a short period. you need to redo the qPCR for 7, 14, and 21 days. It is a little far fetch to conclude that BPA inhibits osteoblast differentiation if your qPCR only is for 24 hrs

But great that you added the primers, most people forget!

Response: Thank you for your comments.

The incubation time used to determine the effect of a treatment on gene expression in osteoblasts in a large number of studies is 24h, which is why it has been the time chosen. Below, we indicate different references to support this fact:

  • Kędzierska et al. Acta Biochim Pol. 2018 Nov 15;65(4):573-571. doi: 10.18388/abp.2018_2633
  • Thent et al.. Life Sci. 2018, 210, 214–223, doi:10.1016/j.lfs.2018.08.057.
  • Lencel et al. Bone. 2011 Feb;48(2):242-9. doi: 10.1016/j.bone.2010.09.001.
  • Kinoshita et al. Sci Rep. 2021 Oct 13;11(1):20360. doi: 10.1038/s41598-021-00034-x.
  • Štefančík M, Válková L, Veverková J, Balvan J, Vičar T, Babula P, Mašek J, Kulich P, Štefančík et al. Environ Sci Pollut Res Int. 2021 Feb;28(5):6018-6029. doi: 10.1007/s11356-020-10908-8.

On the other hand, studying gene expression in osteoblasts under mineralization conditions (osteogenic medium and high cell density), as you suggest could certainly be interesting. We will keep in mind for future studies, since the three human osteoblast lines established for this study, being primary, have a limited number of passages.

  1. Please make sure that you have the molarities in superscript doses of 10-5, 10-6, or 10-7 A not "doses of 10-5, 10-6, or 10-7 M. A”

Response: Thank you for your review and sorry for the errors, which have been corrected throughout the text.

  1. Please revise sentence "Finally, three primary osteo-208 blast cell lines were established." I guess cell lines is wrong here since you used primary human cells.

Response: The sentence is correct, since a line of primary osteoblasts is obtained from each sample. In studies of these characteristics, we can work either with typed cell lines from collections such as ATCC or with cell lines obtained by primary culture from human donors, as in our case.

  1. Please state that you Statistical analysis was done with the control as control. However it also be interesting to know if you had sign diff between he different BPA concentrations

Response: We have also included your suggestion, which helps to improve the understanding of the data: All data were analysed in relation to the control group.

Regarding your suggestion about comparison between different concentrations, it is an interesting analysis and we will keep it in mind for future studies.

Reviewer 2 Report

Line 39: correction needed in ”containers or utensils ils [7,8].”

Abstract, Line 17. Concentrations should be written using superscripts.

There are many studies published on the effect of BPA on osteoblasts. How do the authors justify the Novelty of this work?

Is there any specific reason why authors did not consider checking the effect of BPA (performing the same tests) on osteoclasts?

The Introduction is very short. There should be more explanation and background of BPA.

Similarly, conclusion too is very brief. It deserves more elaboration.

Author Response

Reviewer 2

Line 39: correction needed in ”containers or utensils ils [7,8].”

Abstract, Line 17. Concentrations should be written using superscripts.

Response: Thank you for your review and sorry for the mistakes, which have been corrected.

There are many studies published on the effect of BPA on osteoblasts. How do the authors justify the Novelty of this work?

Response: To our knowledge this is the first article showing the effect of BPA on human osteoblasts. The few studies to date have used typed cell lines such as MC3T3-E1 mice line and the hFOB1.19 line of human fetal nature. On the other hand, our study analyzes a greater number of parameters: growth (proliferation and induction of apoptosis/necrosis), alkaline phosphatase synthesis, mineralization (not previously studied), phagocytosis (not previously studied), antigenic expression (not previously studied) and gene expression (with a greater number of osteogenic genes). Therefore, we consider that our study is the first work in human cultured osteoblasts, which also displays a large number of physiological parameters of this cell population.

Is there any specific reason why authors did not consider checking the effect of BPA (performing the same tests) on osteoclasts?

Response: Thank you for your suggestion. Osteoblasts and osteoclasts are completely different cell populations (different isolation and characterization techniques) and with clearly different functions at the bone level which implies the analysis of completely different parameters. The articles referring to both populations are generally review articles and very limited, among which the following are worth mentioning:

  • Giannattasio et al. Endocr Metab Immune Disord Drug Targets. 2021 Jan 18. doi: 10.2174/1871530321666210118163538.
  • Yaglova & Yaglov. Sovrem Tekhnologii Med. 2021;13(2):84-94. doi: 10.17691/stm2021.13.2.10.
  • Thent et al. Life Sci. 2018 Apr 1;198:1-7. doi: 10.1016/j.lfs.2018.02.013.

The Introduction is very short. There should be more explanation and background of BPA. Similarly, conclusion too is very brief. It deserves more elaboration.

Response: Following your suggestion, the introduction and conclusion have been modified. We have also included some new references.

Introduction:

Line 29-34: “Bisphenol A (BPA), or 4,4′-(Propane-2,2-diyl) diphenol, is a chemical compound used as a major component in the manufacture of polycarbonate plastics, epoxy resins and other polymeric materials, as well as in certain paper products. This material is employed in the manufacture of products of daily use in the home, including food containers, bottles, utensils, CDs, pens and toys, among others. They are also used as a coating for food cans, water tanks, as an adjuvant in dental treatments or as a material for medical devices”.

Line 46-63: “In addition, BPA has demonstrated the ability to interact with androgen, glucocorticoid and thyroid hormone receptors [10]. In terms of pharmacokinetics, after absorption, BPA is rapidly metabolized to several inactive metabolites and free BPA is excreted mainly in the urine. Free BPA has been detected in the urine of adults and children, as well as in the serum of pregnant women, umbilical cord serum and breast milk [10]. Thus, it has the ability to cross the blood-placental barrier and affect the intrauterine development of embryos and fetuses [11].

BPA is involved in the regulation of cancer cell proliferation, migration, invasion and apoptosis. It is also implicated in anti-cancer drug resistance through several signalling pathways activated by BPA binding to nuclear and membrane receptors such as ERα/β/γ, androgen receptor, insulin-like growth factor-1 receptor (IGF-1R), among others [10]. Interaction with these receptors also appears to be responsible for BPA toxicity on the reproductive system, interfering with breast formation, germ cell maturation or placental adhesion to the endometrium. It could also affect to inflammatory and immune responses by disrupting different cell signaling pathways mediated by cytokines and various immune cells, such as T and B lymphocytes, macrophages, mast cells, natural killers or dendritic cells [12].

Conclusions

Line 355-357: …..the home [32,38]. Our results contribute to explain the increase in the prevalence of bone pathologies, such as osteoporosis, as a consequence of the high exposure of the population to this endocrine disruptor. However, further in vitro and…..

  1. Murata, M.; Kang, J.-H. Bisphenol A (BPA) and Cell Signaling Pathways. Biotechnol Adv 2018, 36, 311–327, doi:10.1016/j.biotechadv.2017.12.002.
  2. Yaglova, N.V.; Yaglov, V.V. Endocrine Disruptors as a New Etiologic Factor of Bone Tissue Diseases (Review). Sovrem Tekhnologii Med 2021, 13, 84–94, doi:10.17691/stm2021.13.2.10.
  3. Araiza, V.H.D.R.; Mendoza, M.S.; Castro, K.E.N.; Cruz, S.M.; Rueda, K.C.; de Leon, C.T.G.; Morales Montor, J. Bisphenol A, an Endocrine-Disruptor Compund, That Modulates the Immune Response to Infections. Front Biosci (Landmark Ed) 2021, 26, 346–362, doi:10.2741/4897.

Reviewer 3 Report

The introduction is very brief and requires a much better structure

Please use material and methods as section 2 !

The English should be slightly improved

Some quantitative details are required in conclusion

Some more novel references are helpful

Author Response

Reviewer 3

Thank you for all your comments and for contributing to enrich the manuscript.

  1. The introduction is very brief and requires a much better structure

Response: Following your suggestion, the introduction have been modified. We have also included some new references.

Introduction:

Line 29-34: “Bisphenol A (BPA), or 4,4′-(Propane-2,2-diyl) diphenol, is a chemical compound used as a major component in the manufacture of polycarbonate plastics, epoxy resins and other polymeric materials, as well as in certain paper products. This material is employed in the manufacture of products of daily use in the home, including food containers, bottles, utensils, CDs, pens and toys, among others. They are also used as a coating for food cans, water tanks, as an adjuvant in dental treatments or as a material for medical devices”.

Line 46-63: “In addition, BPA has demonstrated the ability to interact with androgen, glucocorticoid and thyroid hormone receptors [10]. In terms of pharmacokinetics, after absorption, BPA is rapidly metabolized to several inactive metabolites and free BPA is excreted mainly in the urine. Free BPA has been detected in the urine of adults and children, as well as in the serum of pregnant women, umbilical cord serum and breast milk [10]. Thus, it has the ability to cross the blood-placental barrier and affect the intrauterine development of embryos and fetuses [11].

BPA is involved in the regulation of cancer cell proliferation, migration, invasion and apoptosis. It is also implicated in anti-cancer drug resistance through several signalling pathways activated by BPA binding to nuclear and membrane receptors such as ERα/β/γ, androgen receptor, insulin-like growth factor-1 receptor (IGF-1R), among others [10]. Interaction with these receptors also appears to be responsible for BPA toxicity on the reproductive system, interfering with breast formation, germ cell maturation or placental adhesion to the endometrium. It could also affect to inflammatory and immune responses by disrupting different cell signaling pathways mediated by cytokines and various immune cells, such as T and B lymphocytes, macrophages, mast cells, natural killers or dendritic cells [12].

  1. Murata, M.; Kang, J.-H. Bisphenol A (BPA) and Cell Signaling Pathways. Biotechnol Adv 2018, 36, 311–327, doi:10.1016/j.biotechadv.2017.12.002.
  2. Yaglova, N.V.; Yaglov, V.V. Endocrine Disruptors as a New Etiologic Factor of Bone Tissue Diseases (Review). Sovrem Tekhnologii Med 2021, 13, 84–94, doi:10.17691/stm2021.13.2.10.
  3. Araiza, V.H.D.R.; Mendoza, M.S.; Castro, K.E.N.; Cruz, S.M.; Rueda, K.C.; de Leon, C.T.G.; Morales Montor, J. Bisphenol A, an Endocrine-Disruptor Compund, That Modulates the Immune Response to Infections. Front Biosci (Landmark Ed) 2021, 26, 346–362, doi:10.2741/4897.
  4. Please use material and methods as section 2 !

Response: The guideline for authors of this journal include the Materials and Methods after the Discussion, so it should be section 4.

  1. The English should be slightly improved

Response: Thank you for your suggestion. The manuscript has been revised by a native translator.

  1. Some quantitative details are required in conclusion

Response: Following your suggestion we have included the significant changes produced by the treatment.

Line 350-351: These in vitro findings confirm the significant adverse effects of BPA on human osteoblasts.

  1. Some more novel references are helpful

Response: We have included new citations as suggested by the reviewer, although specific original articles analyzing the effect of Bisphenol A on osteoblasts are lacking as of 2018. 

  1. Murata, M.; Kang, J.-H. Bisphenol A (BPA) and Cell Signaling Pathways. Biotechnol Adv 2018, 36, 311–327, doi:10.1016/j.biotechadv.2017.12.002.
  2. Yaglova, N.V.; Yaglov, V.V. Endocrine Disruptors as a New Etiologic Factor of Bone Tissue Diseases (Review). Sovrem Tekhnologii Med 2021, 13, 84–94, doi:10.17691/stm2021.13.2.10.
  3. Araiza, V.H.D.R.; Mendoza, M.S.; Castro, K.E.N.; Cruz, S.M.; Rueda, K.C.; de Leon, C.T.G.; Morales Montor, J. Bisphenol A, an Endocrine-Disruptor Compund, That Modulates the Immune Response to Infections. Front Biosci (Landmark Ed) 2021, 26, 346–362, doi:10.2741/4897.
  4. Miki, Y.; Hata, S.; Nagasaki, S.; Suzuki, T.; Ito, K.; Kumamoto, H.; Sasano, H. Steroid and Xenobiotic Receptor-Mediated Effects of Bisphenol A on Human Osteoblasts. Life Sci 2016, 155, 29–35, doi:10.1016/j.lfs.2016.05.013.

Round 2

Reviewer 1 Report

no further comments

Reviewer 3 Report

,